# Capturing the motion of every joint: 3D human pose and shape estimation with independent tokens

**Sen Yang**[1,2][*] **Wen Heng**[3]**, Gang Liu**[3]**, Guozhong Luo**[3]**, Wankou Yang**[1,2][†]**, Gang Yu**[3]
[1]School of Automation, Southeast University, China    [2]Key Lab of Measurement and Control of Complex Systems of Engineering, Ministry of Education, Southeast University, Nanjing, China
[3]Tencent PCG
{yangsenius,wkyang}@seu.edu.cn
{wenheng,sylvainliu,alexantaluo,skicyyu}@tencent.com

## Abstract

In this paper we present a novel method to estimate 3D human pose and shape from monocular videos. This task requires directly recovering pixel-alignment 3D human pose and body shape from monocular images or videos, which is challenging due to its inherent ambiguity. To improve precision, existing methods highly rely on the initialized mean pose and shape as prior estimates and parameter regression with an iterative error feedback manner. In addition, video-based approaches model the overall change over the image-level features to temporally enhance the single-frame feature, but fail to capture the rotational motion at the joint level, and cannot guarantee local temporal consistency. To address these issues, we propose a novel Transformer-based model with a design of independent tokens. First, we introduce three types of tokens independent of the image feature: *joint rotation tokens, shape token, and camera token*. By progressively interacting with image features through Transformer layers, these tokens learn to encode the prior knowledge of human 3D joint rotations, body shape, and position information from large-scale data, and are updated to estimate SMPL parameters conditioned on a given image. Second, benefiting from the proposed token-based representation, we further use a temporal model to focus on capturing the rotational temporal information of each joint, which is empirically conducive to preventing large jitters in local parts. Despite being conceptually simple, the proposed method attains superior performances on the 3DPW and Human3.6M datasets. Using ResNet-50 and Transformer architectures, it obtains 42.0 mm error on the PA-MPJPE metric of the challenging 3DPW, outperforming state-of-the-art counterparts by a large margin. Code will be publicly available[1].

## 1 Introduction

Capturing the motion of the human body pose has great values in widespread applications, such as movement analysis, human-computer interaction, films making, digital avatar animation, and virtual reality. Traditional marker-based motion capture system can acquire accurate movement information of humans, but is only applicable to limited scenes due to the time-consuming fitting process and prohibitively expensive costs. In contrast, markerless motion capture based on RGB image and video processing algorithms is a promising alternative that has attracted numerous research in the fields of deep learning and computer vision. Especially, thanks to the parameteric SMPL model (Loper et al., 2015) and various diverse datasets with 3D annotations (Ionescu et al., 2013; Mehta et al., 2017; von Marcard et al., 2018), remarkable progress has been made on monocular 3D human pose and shape estimation and motion capture.

---

[*]This work was done when Sen Yang was intern at Tecent PCG.

[†]Corresponding Author

[1]https://github.com/yangsenius/INT_HMR_Model

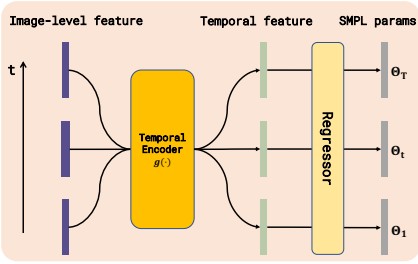 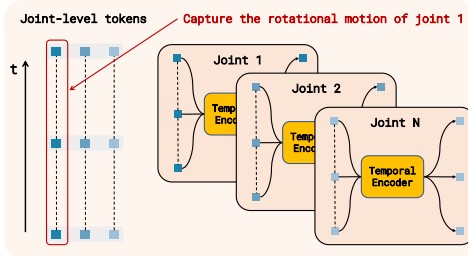

Figure 1: **Left:** Mainstream temporal-based human mesh methods, e.g. (Kanazawa et al., 2019a; Kocabas et al., 2020; Choi et al., 2021), adopt a temporal encoder to mix temporal information from past and future frames and then regress the SMPL parameters from the *temporally enhanced feature* for each frame. **Right:** Our method first acquires tokens of each joint in the time dimension and then separately capture the motion of each joint using a shared temporal encoder.

Existing regression-based human mesh recovery methods are actually implicitly based on an assumption that predicting 3d body joint rotations and human shape strongly depends on the given image features. The pose and shape parameters are directly estimated from the image feature using MLP regressors. Nevertheless, due to the inherent ambiguity, the mapping from the 2D image feature to 3D pose and shape is an ill-posed problem. To achieve accurate pose and shape estimation, it is necessary to initialize the mean pose and shape parameters and use an *iterative residual regression* manner to reduce error. Such an end-to-end learning and inference scheme (Kanazawa et al., 2018) has been proven to be effective in practice, but ignores the temporal information and produces implausible human motions and unsatisfactory pose jitters for video streaming data. Video-based methods such as (Kanazawa et al., 2019a; Kocabas et al., 2020; Choi et al., 2021; Wei et al., 2022) may leverage large-scale motion capture data as priors and exploit temporal information among different frames to penalize implausible motions. They usually enhance singe-frame feature using a temporal encoder and then still use a deep regressor to predict SMPL parameters based on the temporally enhanced image feature, as shown in the left subfigure of Fig.1. This scheme, however, is unable to focus on joint-level rotational motion specific to each joint, failing to ensure the temporal consistency of local joints. To address these problems, we attempt to understand the human 3D reconstruction from a causal perspective. We argue that assuming a still background, the primary causes behind the image pixel changes and human body appearance changes are 1) the motions of 3D joint rotations in human skeletal dynamics and 2) the viewpoint changes of the observer (camera). In fact, a prior human body model exists independently of a given specific image. And the 3D relative rotations of all joints (relative to the parent joint) and body shape can be abstracted beyond image pixels and independent of the image contents and observer views. In other words, the joint rotations cannot be "seen" and they are image-independent and viewpoint-independent concepts.

Based on the considerations above, we propose a novel 3D human pose and shape estimation model based on **in**dependent **t**okens (INT). The core idea of the model is to introduce three types of independent tokens that specifically encode the 3D rotation information of every joint, the shape of human body and the information about camera. These initialized tokens learn prior knowledge and mutual relationships from large-scale training data, requiring neither an iterative regressor to take mean shape and pose as initial estimate (Kanazawa et al., 2018; Kolotouros et al., 2019a; Kocabas et al., 2020; Choi et al., 2021), nor a kinematic topology decoder defined by human prior knowledge (Wan et al., 2021) . Given an image as a conditional observation, these tokens are repeatedly updated by interacting with 2D image evidence using a Transformer (Vaswani et al., 2017). Finally, they are transformed into the posterior estimates of pose, shape and camera parameters. As a consequence, this method of abstracting joint rotation tokens from image pixels can represent the motion state of each joint and establish correlations in time dimension. Benefiting from this, we can separately capture the temporal rotational motion of every joint by sending the tokens of each joint at different timestamps to a temporal model. In comparison to capturing the overall temporal changes in image features and the whole pose, this modeling scheme focuses on capturing separate rotational motions of all joints, which is conducive to maintaining the temporal coherence and rationality of each joint rotation.

We evaluate our model on the challenging 3DPW (von Marcard et al., 2018) benchmark and Human3.6m (Ionescu et al., 2013). Using vanilla ResNet-50 and Transformer architectures, our model obtains 42.0 mm error in PA-MPJPE metric for 3DPW, outperforming all state-of-the-art counterparts with a large margin. The same model obtains 38.4 mm error in PA-MPJPE metric for Human3.6m, which is on par with the state-of-the-art methods. Also, the qualitative results show that our model produces accurate pixel-alignment human mesh reconstructions for indoor or in-the-wild images, and shows fewer motion jitters in local joints when processing video data. We strongly encourage the readers to see the video results in the supplementary materials for reference and comparison.

## 2 RELATED WORK

Great progress has been made in the monocular image and video based 3D human pose and shape estimation, thanks to the parametric human mesh models (Loper et al., 2015; Pavlakos et al., 2019; Joo et al., 2018), particularly the SMPL (Loper et al., 2015) and SMPL-X (Pavlakos et al., 2019) models. The mainstream parametric methods are usually can be classified as two categories: optimization-based and regression-based. SMPLify (Bogo et al., 2016) is the first automatic optimization-based approach. It fits SMPL to 2D detected keypoint, using strong data priors to optimize the SMPL parameters. SPIN (Kolotouros et al., 2019a) proposes fitting within the training loop to produce pixel-accurate fittings, where the fittings are used in training instead of in test time. There are also methods further using human silhouette (Lassner et al., 2017) or multi-views information (Huang et al., 2017) to accomplish the optimization.

Regression-based scheme has recently received extensive research (Kanazawa et al., 2018, 2019a; Kocabas et al., 2020; Sun et al., 2019b; Doersch & Zisserman, 2019; Choi et al., 2021; Lin et al., 2021b,a), due to its directness and effectiveness. HMR (Kanazawa et al., 2018), is the representative regression-based methods, using an image encoder and regressor to predict the pose, shape and camera parameters. To train the model well and make sure the realistic of the pose and shape, the reprojection loss and adversarial loss are introduced to leverage unpaired 2D-to-3D supervision. In addition, several non-parameteric mesh regression methods are proposed to directly regress the mesh vertices coordinates, including Pose2Mesh (Choi et al., 2020), Convolution Mesh Regression (Kolotouros et al., 2019b), I2L-MeshNet (Moon & Lee, 2020), and the Transformer-based METRO (Lin et al., 2021b) and Mesh Graphormer (Lin et al., 2021a).

Beyond estimating pose and shape from a singe image, video-based methods consider to fully dig the temporal motion information hidden in video data to improve the accuracy and robustness. HMMR (Kanazawa et al., 2019a) learns the human dynamics to predict pose and shape for past and future frames. VIBE (Kocabas et al., 2020) encodes temporal feature using a GRU and adopts an adversarial learning framework to learn kinematically plausible motion from a large-scale motion capture dataset. TCMR (Choi et al., 2021) introduces PoseForecast to forecast additional temporal features from past and future frames without a current frame. MAED (Wan et al., 2021) proposes to use a spatial-temporal encoder to learn temporally enhanced image features and regress the joint rotations following a defined kinematic topology. In contrast to these methods, our goal is to encode joint-level features, shape and camera information separately, rather than encoding all the information into a unified image feature vector. Since we use independent tokens to encode the rotational information of each joint, we can model the *inner temporal patterns* when each joint rotates over time. Compared with MAED, we make no assumptions about the *directed* dependencies between rotations of joints, because its tree-based topology fails to capture important dependencies between *non-adjacent* joints. In our modeling scheme, the joint tokens can freely learn *undirected* relationships between any pairs of joints from large-scale data and a given image.

Transformer (Vaswani et al., 2017) is proposed as a powerful model that is suitable for sequence-to-sequence modeling. Transformer has less inductive bias and shows powerful performance when trained with sufficient data. It is applied to various vision tasks including image classification (Dosovitskiy et al., 2020; Touvron et al., 2021; Liu et al., 2021), object detection (Carion et al., 2020; Chen et al., 2021), segmentation (Wang et al., 2021; Xie et al., 2021), video classification (Arnab et al., 2021), 2D/3D human pose estimation (Yang et al., 2021a; Li et al., 2021c,b; Yuan et al., 2021; Yang et al., 2021b; Mao et al., 2022; Zheng et al., 2021) and 3D human mesh reconstruction (Lin et al., 2021b,a; Wan et al., 2021), etc. In this work, inspired by the token-based Transformer designs (Devlin et al., 2018; Dosovitskiy et al., 2020; Li et al., 2021c), we use multiple independent tokens to represent

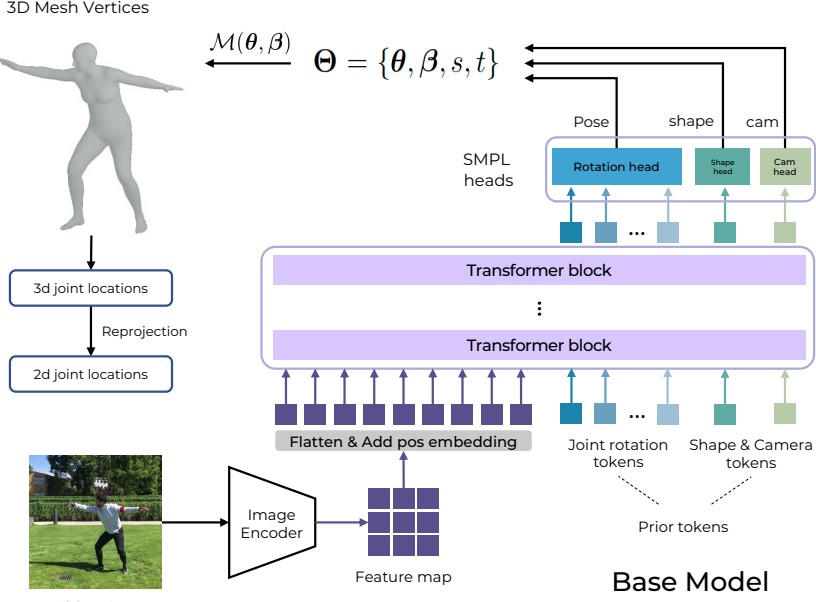

Figure 2: Base model for singe-frame input. We first use an image encoder to extract feature maps from a given cropped image, then we flatten the feature map into a sequence and add a learnable position embedding. The learnable joint rotation tokens, shape and camera tokens are appended to the sequence and sent to the transformer. Finally, we use three linear heads, called rotation head, shape head and camera head, to convert the joint rotation tokens, shape token and camera token to the SMPL parameters, for 3D mesh reconstruction and 2D reprojection on the image plane.

the information with respect to joint 3D rotation, body shape and camera parameter. And we also use Transformer to conduct sequence-to-sequence temporal modeling.

## 3 METHOD

Our goal is to build a model that represents joint rotations, shape and camera information using tokens independent of image feature and further captures the rotational motion information of each joint from video data. In this section, we first revisit prior SMPL-based human mesh recovery methods and then describe our model design.

### 3.1 REVISITING SMPL-BASED HUMAN MESH RECOVERY

The classic human mesh recovery (HMR) methods (Kanazawa et al., 2018, 2019a; Kocabas et al., 2020) represent human body as a mesh using the parameteric SMPL (Loper et al., 2015) model. The SMPL mesh model is a differentiable function that output 6890 surface vertices $\mathcal{M}(\boldsymbol{\theta}, \boldsymbol{\beta}) \in \mathbb{R}^{6890 \times 3}$, which are deformed with linear blend skinning driven by the pose $\boldsymbol{\theta} \in \mathbb{R}^{72}$ and shape $\boldsymbol{\beta} \in \mathbb{R}^{10}$ parameters. The pose $\boldsymbol{\theta}$ parameter include the global rotation $R$ and 23 relative joint rotations in axis-angle format. To obtain the 3D positions of body joints, a pretrained linear regressor $W$ is used to achieve $J_{3d} = W\mathcal{M}(\boldsymbol{\theta}, \boldsymbol{\beta})$. To leverage 2D joint supervision, a weak-perspective camera model is usually used to project 3D joint positions into the 2D image plane, i.e., $J_{2d} = s\Pi(RJ_{3d}) + \boldsymbol{t}$, where $\Pi$ is an orthographic projection, the scale value $s$ and translation $\boldsymbol{t} \in \mathbb{R}^2$ are camera related parameters.

For the image-based HMR methods like (Kanazawa et al., 2018; Kolotouros et al., 2019a), an image encoder $f(\cdot)$ and a MLP regressor are used to estimate the set of reconstruction parameters $\boldsymbol{\Theta} = \{\boldsymbol{\theta}, \boldsymbol{\beta}, s, t\}$, which constitutes an 85-dim vector to regress. These parameters are iteratively regressed from the encoded image feature vector $\boldsymbol{f}$ by the regressor. For the video-based HMR methods like (Kocabas et al., 2020; Choi et al., 2021; Kanazawa et al., 2019a; Wan et al., 2021; Wei et al., 2022), temporal models based on 1D convolution (Kanazawa et al., 2019a), GRU (Choi et al.,

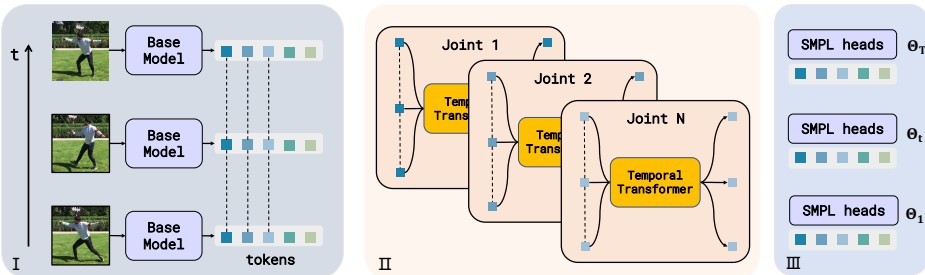

Figure 3: The overall temporal model framework. **I. The base model**. We feed the frames of a given video clip to the same base model and achieve the tokens for each frame. **II. The temporal model**. We use a transformer as the rotation motion encoder to capture the motion of each joint. **III. The SMPL heads**. We feed the updated joint tokens, shape token and camera token of each frame to the SMPL heads shared with image-based model, to achieve the final SMPL parameters.

2021) and self-attention models (Kocabas et al., 2020; Wan et al., 2021) are introduced to capture the motion information in consecutive video frames. A temporal encoder $g(\cdot)$ is exploited to achieve temporally encoded feature vector, formulated as a process like: $\boldsymbol{m}_t = g(\boldsymbol{f}_{t-T/2}, ..., \boldsymbol{f}_t, ..., \boldsymbol{f}_{t+T/2})$. Also, a regressor is used to estimate the $\boldsymbol{\Theta}_t$ from the current frame's feature $\boldsymbol{m}_t$ encoded with temporal information.

## 3.2 ESTIMATING SMPL PARAMETERS BASED ON INDEPENDENT TOKENS

In this section, we first introduce the token based semantic representation and then describe our model design for single frame input, mainly including the `Base model` and `SMPL heads`.

**Joint rotation tokens, shape token & camera token.** We introduce three types of token representations: (1) joint rotation tokens consist of 24 tokens, each of which encodes the joint 3D relative rotation information (including the global rotation) $\boldsymbol{r}_i \in \mathbb{R}^d, i = 1, .., 24$; (2) shape token is a token vector $\boldsymbol{s} \in \mathbb{R}^d$ encoding the body shape information; (3) camera token is also a token vector $\boldsymbol{c} \in \mathbb{R}^d$ encoding the translation and scale information. $d$ is the vector dimension for all tokens.

**Base model.** Inspired by ViT (Dosovitskiy et al., 2020) and TokenPose (Li et al., 2021c), we embody our scheme into a Transformer-based architecture design (Fig. 2). We adopt a CNN to extract image feature map $\boldsymbol{f} \in \mathbb{R}^{c \times h \times w}$ from a given RGB image $I$ cropped with a human body. We reshape the extracted feature map into a sequence of flattened patches and apply patch embedding $\mathbf{E}$ (linear transformation) to each patch to achieve the sequence $\boldsymbol{f}_p \in \mathbb{R}^{S \times d}$ where $S = h \times w$. We append the totally learnable joint rotation tokens, shape token and camera token to the sequence $\boldsymbol{f}_p$, namely *prior tokens*. We only inject the learnable position embedding $\mathbf{PE} \in \mathbb{R}^{S \times d}$ into the $\boldsymbol{f}_p$ to preserve the 2D structure position information. Then we send the whole sequence $\boldsymbol{S}_0 \in \mathbb{R}^{(S+24+1+1) \times d}$ to a standard Transformer encoder with $L$ layers and achieve these corresponding tokens from the final layer.

**SMPL heads.** To achieve the estimated SMPL parameters $\Theta = \{\boldsymbol{\theta}, \boldsymbol{\beta}, s, t\}$ for 3D human mesh reconstruction, we use three *linear* SMPL heads – *rotation head*, *shape head* and *camera head* – to transform the corresponding tokens outputted from the final transformer layer. Particularly, the rotation head (a shared linear layer) transforms each joint token into a 6D rotation representation (Zhou et al., 2019). The shape and camera heads (two linear layers) separately transform the shape token and camera token into the shape parameters (10-dim vector) and camera parameters (3-dim vector). Finally, we convert the 6D rotation representations to SMPL pose (in axis-angle format) and use these parameters to generate the human mesh. Further, we obtain the predicted joint 3D locations $\boldsymbol{J}_{3d}$ and then project them into 2D locations $\boldsymbol{J}_{2d}$ using a weak-perspective camera model.

## 3.3 ROTATIONAL MOTION CAPTURING USING A TEMPORAL TRANSFORMER

We aim to capture the rotational motion at the joint level. Given a video clip $V = \{I_t\}_{t=1}^T$ of length $T$, we feed these $T$ frames to the Base model and acquire the estimated $N$ joint rotation tokens for each frame: $\{\hat{\boldsymbol{r}}_1^t, ..., \hat{\boldsymbol{r}}_N^t\}_{t=1}^T \in \mathbb{R}^{T \times N \times d}$, from the final transformer layer.

We use another standard Transformer as the temporal model to capture the motion of each joint; we denote it as `Temporal Transformer`. For the $n$-th type joint token $\boldsymbol{r}_n$, such as the left knee, the token sequence formed in the time axis is $X_n = \{\hat{\boldsymbol{r}}_n^1, \hat{\boldsymbol{r}}_n^2, ..., \hat{\boldsymbol{r}}_n^T\} \in \mathbb{R}^{T \times d}$, where $n \in \{1, ..., N\}$. We feed each sequence $X_n$ to the Temporal Transformer and achieve a new sequence $X_n'$, so that each updated joint token from a particular moment is mixed with the joint rotation information from past and future frames. Then, we reshape the temporally updated tokens $\{X_1', ..., X_N'\}$ from $N \times T \times d$ to $T \times N \times d$. Note that the temporal information in camera and shape tokens is not taken into consideration in our model since we hope to capture the pure rotational motion information of joints. Finally, for timestamp $t$, the updated $N$ joint tokens are then fed into the rotation head to achieve the joint rotations $\boldsymbol{\theta}_t$; the shape token and camera token outputted from the Base model are fed to the shape head and camera head to achieve the $\boldsymbol{\beta}_t, s_t, \boldsymbol{t}_t$. The overall framework is shown in Fig. 3.

## 3.4 Loss function

Leveraging full supervision from different formats of annotations is critical to train the model well and attain the generalization in different cases. Following common human mesh recovery methods, we use SMPL parameters loss, L2 normalization, 3D joint location loss and projected 2D joint location loss, when the corresponding SMPL, 3D/2D location supervision signals are available.

$$\mathcal{L}_{smpl} = w_\theta \cdot \|\boldsymbol{\theta} - \boldsymbol{\theta}_{gt}\|_2 + w_\beta \cdot \|\boldsymbol{\beta} - \boldsymbol{\beta}_{gt}\|_2 ,$$

$$\mathcal{L}_{norm} = \|\boldsymbol{\theta}\|_2 + \|\boldsymbol{\beta}\|_2 ,$$

$$\mathcal{L}_{3D} = \|\boldsymbol{J}_{3d} - \boldsymbol{J}_{3dgt}\|_2 , \mathcal{L}_{2D} = \|\boldsymbol{J}_{2d} - \boldsymbol{J}_{2dgt}\|_2 ,$$

$$\mathcal{L}_{temp} = \left\|(\boldsymbol{J}_{3d}^{t+1} - \boldsymbol{J}_{3d}^t) - (\boldsymbol{J}_{3dgt}^{t+1} - \boldsymbol{J}_{3dgt}^t)\right\|_2 ,$$

$$\mathcal{L} = \mathcal{L}_{smpl} + w_{norm} \cdot \mathcal{L}_{norm} + w_{3D} \cdot \mathcal{L}_{3D} + w_{2D} \cdot \mathcal{L}_{2D} + w_{temp} \cdot \mathcal{L}_{temp}.$$

$\boldsymbol{\theta}_{gt}, \boldsymbol{\beta}_{gt}$ are the groundtruth SMPL parameters. $\boldsymbol{J}_{3D}, \boldsymbol{J}_{2D}$ are the groundtruth joint 3D and 2D locations. The $\mathcal{L}_{temp}$ is a temporal loss for video data, which supervises the velocity of the joint temporal movement in 3D space. The $w_\theta, w_\beta, w_{norm}, w_{3D}, w_{2D}, w_{temp}$ are the weights to balance all of loss functions (see more details in Appendix C).

## 4 Experiments

**Training data & Model setups.** Following (Kanazawa et al., 2018; Kocabas et al., 2020; Wan et al., 2021), we use mixed 3D video, 2D video and 2D image datasets for training. The details of model configurations and training data are described in Appendix A and Appendix B.

**Evaluation.** We report results on 3DPW and Human3.6m datasets, using 4 standard metrics, including Procrustes-Aligned Mean Per Joint Position Error (PA-MPJPE), Mean Per Joint Position Error (MPJPE), Per Vertex Error (PVE) and ACCELeration error (ACCEL). The unit for these metrics is millimeter (mm). We evaluate the models trained w/ and wo/ 3DPW training set for comparison with previous methods.

**Progressive training scheme.** In practice, we find how to exploit these training datasets with such multi-modal supervision is critical to training the whole model well and ensuring the generalization to in-the-wild scenes. Following MAED (Wan et al., 2021), we develop an improved progressive training scheme adapting to our model, which consists of three training phases. Please see more details about this training scheme in Appendix C.

### 4.1 Comparison with state-of-the-art methods

**Quantitative results.** In Tab. 4, we compare our method with state-of-the-art image-based and video-based HMR methods. We evaluate the models trained w/ and w/o 3DPW training set for fair comparisons. For 3DPW, our model substantially outperforms all these methods, mainly in PA-MPJPE, MPJPE and PVE. For Human3.6m, our model is on par with state-of-the-art methods. MAED (Wan et al., 2021) is our main competitor, as both use the same backbone and training data. The results show that our model achieve superior performances in all metrics over MAED, particularly in PA-MPJPE on 3DPW gaining 8.1% performance improvement. Our model achieves a

Table 1: Comparisons to state-of-the-art models on 3DPW and Human3.6M datasets. INT-1 model is trained with first two phases, and separately evaluated the best performance for both datasets. INT-2 model is trained with three phases, and the best model for 3DPW is directly evaluated on Human3.6m. † represents training w/o 3DPW training dataset. ∗ represents training w/ 3DPW training dataset. For fair comparison, we also list the CNN backbones (ResNet (He et al., 2016) or HRNet (Sun et al., 2019a)) used in different methods.

| | Models | Backbone | 3DPW | | | | H36M | |
| | | | PA-MPJPE ↓ | MPJPE ↓ | PVE ↓ | Accel ↓ | PA-MPJPE ↓ | MPJPE ↓ |
|---|---|---|---|---|---|---|---|---|
| Image-based | HMR† (Kanazawa et al., 2018) | ResNet-50 | 76.7 | 130.0 | - | 37.4 | 56.8 | 88 |
| | Neural Body† (Omran et al., 2018) | ResNet-101 | - | - | - | - | 59.9 | - |
| | Mesh Regression† (Kolotouros et al., 2019b) | ResNet-50 | 70.2 | - | - | - | 50.1 | - |
| | SPIN† (Kolotouros et al., 2019a) | ResNet-50 | 59.2 | 96.9 | 116.4 | 29.8 | 41.1 | - |
| | I2l-MeshNet† (Moon & Lee, 2020) | ResNet-50 | 57.7 | 93.2 | 110.1 | 30.9 | 41.1 | 55.7 |
| | PyMAF† (Zhang et al., 2021) | ResNet-50 | 58.9 | 92.8 | 110.1 | - | 40.5 | 57.7 |
| | Hybrik† (Li et al., 2021a) | ResNet-34 | 48.8 | 80.0 | 94.5 | 34.5 | 54.4 | |
| | ROMP∗ (Sun et al., 2021) | ResNet-50 | 49.7 | 79.7 | 94.7 | - | - | - |
| | ROMP∗ (Sun et al., 2021) | HRNet-W32 | 47.3 | 76.7 | 93.4 | - | - | - |
| | PARE† (Kocabas et al., 2021) | ResNet-50 | 52.3 | 82.9 | 99.7 | - | - | - |
| | PARE† (Kocabas et al., 2021) | HRNet-W32 | 50.9 | 82.0 | 97.9 | - | - | - |
| | PARE∗ (Kocabas et al., 2021) | HRNet-W32 | 46.5 | **74.5** | 88.6 | - | - | - |
| | METRO∗ (Lin et al., 2021b) | ResNet-50 | - | - | - | - | 40.6 | 56.5 |
| | METRO∗ (Lin et al., 2021b) | HRNet-W64 | 47.9 | 77.1 | 88.2 | - | 36.7 | 54.0 |
| | Mesh Graphormer∗ (Lin et al., 2021a) | HRNet-W64 | **45.6** | 74.7 | **87.7** | - | **34.5** | **51.2** |
| Video-based | HMMR† (Kanazawa et al., 2019a) | ResNet-50 | 72.6 | 116.5 | 139.3 | 15.2 | 56.9 | - |
| | Sim2Real† (Doersch & Zisserman, 2019) | ResNet-50 | 74.7 | - | - | - | - | - |
| | Temporal Context† (Arnab et al., 2019) | ResNet (from HMR) | 72.2 | - | - | - | 54.3 | 77.8 |
| | Skeleton-disentangled† (Sun et al., 2019b) | ResNet-50 | 69.5 | - | - | - | 42.4 | 59.1 |
| | VIBE∗ (Kocabas et al., 2020) | ResNet (from SPIN) | 51.9 | 82.9 | 99.1 | 23.4 | 41.4 | 65.6 |
| | TCMR† (Choi et al., 2021) | ResNet (from SPIN) | 55.8 | 95.0 | 111.3 | **6.7** | 41.1 | 62.3 |
| | MPS-Net∗ (Wei et al., 2022) | ResNet (from SPIN) | 52.1 | 84.3 | 99.7 | 7.4 | 47.4 | 69.4 |
| | MAED∗ (Wan et al., 2021) | ResNet-50 | 45.7 | 79.1 | 92.6 | 17.6 | 38.7 | 56.4 |
| | **INT-1 (Ours)†** | ResNet-50 | 49.7 | 90.0 | 105.1 | 23.5 | 39.1 | 57.1 |
| | **INT-2 (Ours)∗** | ResNet-50 | **42.0** | **75.6** | **87.9** | 16.5 | **38.4** | **54.9** |

significant improvement when trained with 3DPW, which indicates that using accurate SMPL pose and shape labels as supervision is critical to improve the generalization to in-the-wild scenes. Notably, METRO (Lin et al., 2021b) and Mesh Graphormer (Lin et al., 2021a) are recent Transformer-based SOTA methods, with a stronger CNN extractor HRNet-W64 (Sun et al., 2019a) as the backbone, showing better performance. In comparison to them, our method shows superior performance in PA-MPJPE on 3DPW and comparable results in MPJPE and PVE, even using ResNet-50 (He et al., 2016) as the backbone. These results demonstrate the effectiveness and superiority of our token-based model design.

**Qualitative results.** To demonstrate qualitative mesh reconstruction results, we select a typical showcase of 3DPW dataset to compare our model with the current video-based SOTA method MAED (Wan et al., 2021), as shown in Fig. 4. We can see that MAED performs well in most frames but still produces some bad fit in hard samples. In contrast, our model produces accurate pixel accurate mesh alignment that better fit to 2D human silhouette, resulting in more natural and smoother motion. This phenomenon suggests that our scheme of separate joint rotation predictions may produce more flexible and adaptive human mesh than regressing the whole pose. Please see more in-the-wild examples in the Appendix D for reference.

## 4.2 ABLATION STUDY

Table 2: Study on the effectiveness of temporal modeling. We evaluate the performances of the image-based model and two types of video-based temporal models.

| Mode | Temporal modeling | PA-MPJPE ↓ | Accel ↓ |
|---|---|---|---|
| Image-based | N/A | 45.6 | 23.5 |
| Video-based | whole-pose temporal learning | 43.9 | 19.1 |
| Video-based | separate joint temporal learning | **42.0** | **16.5** |

**Pure image-based model vs. video-based temporal models.** To study the effectiveness of the temporal modeling, we evaluate the models when trained with and without the Temporal Transformer. When using the Temporal Transformer, we conduct two types of temporal modeling schemes – separate joint temporal learning and whole-pose temporal learning. In Tab. 2, the results show that

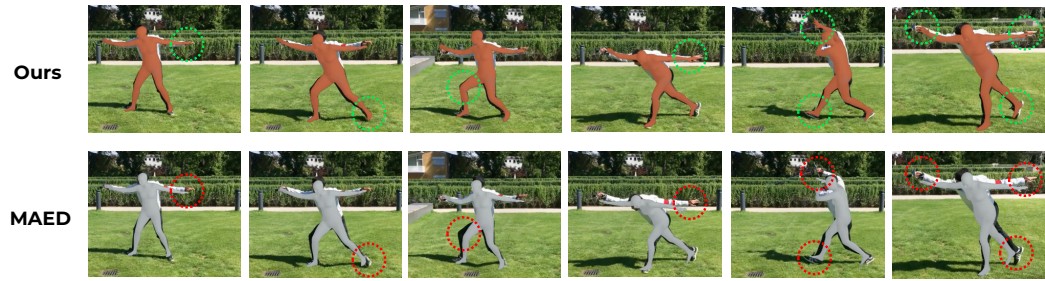

Figure 4: The qualitative comparisons between our model (top) and the reproduced MAED model (Wan et al., 2021) (bottom).

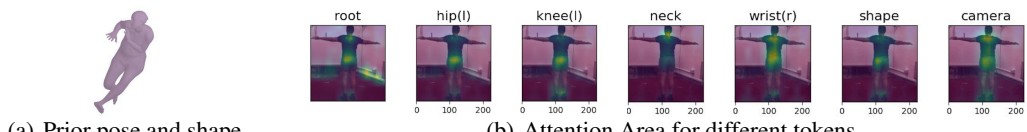

(a) Prior pose and shape        (b) Attention Area for different tokens

Figure 5: Prior knowledge learned in the prior independent tokens. **(a)**: the pose and shape parameters are transformed from the initial joint rotation tokens and shape token by the rotation head and shape head. **(b)**: the corresponding attention areas in an image for some selected initial tokens

video-based ones have obvious advantages in the PA-MPJPE and Accel metrics compared with the image-based one.

**Separate joint temporal learning vs. whole-pose temporal modeling.** We study the differences between the separate joint temporal learning and whole-pose temporal learning. We implement a new model that concatenates all joint tokens (each is $d/24$-dim) together as a pose vector ($d$-dim) and then use the Temporal Transformer to capture the overall changes in pose over time. As shown in Tab. 2, although effective, the whole-pose temporal learning scheme still performs worse than the separate joint temporal learning.

Table 3: Comparisons with the temporal modeling of MAED. We report the MAED result we reproduced.

| Image model | Backbone | Temporal Modeling | PA-MPJPE ↓ | Accel ↓ |
|---|---|---|---|---|
| INT | ViT-base | separate joint temporal learning | 51.6 | 25.3 |
| INT | ResNet50+Transformer | separate joint temporal learnin | **42.0** | **16.5** |
| INT | ResNet50+Transformer | parallel spatial-temporal | 43.0 | 19.4 |
| MAED | ResNet50+Transformer | parallel spatial-temporal | 45.0 | 18.6 |

**Comparison with MAED.** Since we use the same backbone and training data as MAED (Wan et al., 2021), it is relatively fair to compare with the temporal modeling of MAED. We combine the base model with the parallel spatial-temporal mechanism (abbr. parallel). The results in Tab. 3 show that, for PA-MPJPE metric, the parallel scheme performs worse 1 mm than the separate joint temporal learning; but the parallel scheme based INT model still gains 2 mm improvements over MAED.

### 4.3 OBSERVATIONS

Since we explicitly give certain concepts to these learnable tokens, it is desirable to know what information these defined tokens have learned.

**What prior pose and shape are learned?** In Fig. 5(a), we show the prior pose and shape learned in the prior joint rotations and shape token, by transforming them to SMPL parameters by the linear rotation head and shape head. We can see that the prior pose and shape present a reasonable appearance, not a totally random state. Note that we does not leverage any explicit SMPL constraints or image information on these prior joint rotation tokens and shape token. The only learning source

is the backward gradient signals in the training process. This result also indicates that the model learns a *fixed linear transformation* relationship between these joint token vectors and the realistic 3D rotations. In Fig. 5(b), we show the attention areas in an image of for some selected tokens.

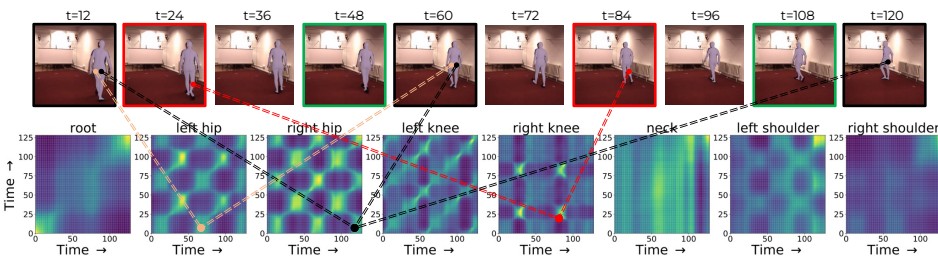

Figure 6: Visualized attention matrices (bottom) between different frames (top) for a video clip ($T = 128$). Colored lines indicate strongly correlated joint rotation tokens between different time points, e.g., for the left hip joint, the joint tokens in the 12-th and 60-th frames have high attention score. Different joints show different rotational temporal patterns.

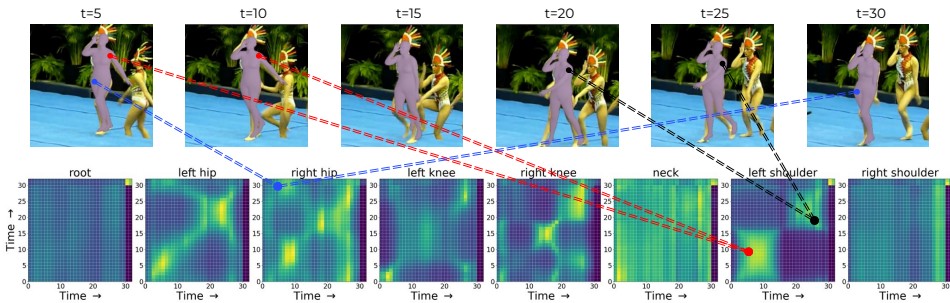

Figure 7: Visualized attention matrices (bottom) between different frames (top) for a clip ($T = 32$). Colored lines indicate strongly correlated joint rotation tokens between different time points.

**Rotational temporal patterns for different joints.** For joint rotation tokens, there is no information exchange between different frames before being sent into the temporal transformer. Through the self-attention interactions in the temporal Transformer, the model builds correlation and integrates the information among different frames. For example, in Fig. 6, the person is walking. For $t = 24$ and $t = 84$ frames, his right knee joint is in the same state of rotation (the right calf is slightly lifted), so the estimates for these two frame should be close. By visualizing the attention matrix in the last layer of temporal transformer, we can see there is indeed a strong correlation between the two frames for the right knee joint token. Similarly, we can observe in Fig. 6 and Fig. 7 that the model captures the periodic joint rotational motions. In this way, the joint angle estimation for the current frame can be corrected and regularized by using the information from past or future frames.

## 5   CONCLUSION

In this paper, we propose a simple yet effective model based on the design of independent tokens to address the problem of 3D human pose and shape estimation. We introduce joint rotation tokens, shape token and camera token to encode the 3D rotations of human joints, body shape and camera parameters for SMPL-based human mesh reconstruction. Thanks to the design of independent tokens, we use a temporal Transformer to capture the temporal motion of each joint separately, which is beneficial for maintaining the temporal rotational coherence of each joint and reducing jitters in local joints. Our model outperforms state-of-the-art counterparts on the challenging 3DPW benchmark and attains comparable results on the Human3.6m. The qualitative results show that our model can produce well-fitted and robust human mesh reconstructions for video data. We also give analysis on what prior information learned in the independent tokens and what temporal patterns are captured by the temporal model. Since we abstract 3D human joints and shape from image pixels as independent concepts/representations, it is possible for future works to incorporate other modal supervision such as text to leverage reasonable constraints on these independent representations for multi-modal learning.

## 6 ACKNOWLEDGEMENT

This work was supported by the National Natural Science Foundation of China under Nos. 62276061, 61773117 and 62006041.

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

# A  MODEL SETUPS

**Base model.** For the Base model, we use a hybrid architecture with ResNet[2] (He et al., 2016) and Transformer[3] (Vaswani et al., 2017), based on the ImageNet pretrained ViT model[4] (Dosovitskiy et al., 2020). The transformer in the Base model has 6 Transformer blocks and the multi-head self-attention layer has 12 attention heads. The embedding dimension is 768 for each token vector and the hidden dimension in the FFN is 3072 (4 times w.r.t. the embedding dimension 768).

**Temporal Transformer.** As we find that increasing the number of layers brings a little improvement but with extra computational overhead, we set the number of layers of Temporal Transformer as 3 for the trade-off. The number of attention heads is also set as 12. The temporal transformer is not pre-trained with any data. We also add a learnable temporal embedding to retain the temporal position information of tokens in the time dimension. In the training, the length $T$ of video clips is 16 sampled at a interval of 8 from the original video data. In inference, we enlarge the bbox with a scale of 1.1 with respect to the original size of bbox.

And in practice, we find setting $T$ to be larger in inference could improve the accuracy, with acceptable additional computational overhead. In Tab. 4, we study on how the input sequence length of the temporal model affect the performances. The results show that longer sequence length brings weak improvements on the overall metrics but stable improvements on the acceleration error. For the INT-2 model, we use $T = 64$ to report the results. Due to the length of learnable temporal embedding is initialized with 16, we interpolate the temporal embedding to the expected length when necessary.

# B  TRAINING DATA

3D video datasets include Human3.6m (Ionescu et al., 2013), MPI-INF-3DHP (Mehta et al., 2017), and 3DPW (von Marcard et al., 2018). 3DPW (von Marcard et al., 2018) is a in-the-wild dataset with accurate pose and shape annotations. We use Human3.6m annotated with pose and shape parameters. PennAction (Zhang et al., 2013) and PoseTrack (Andriluka et al., 2018) are the 2D video datasets with only annotated 2D groundtruth. InstaVariety (Kanazawa et al., 2019b) is the 2D video dataset with pseudo 2D grountruth annotations. For 2D image datasets, we use COCO (Lin et al., 2014), MPII (Andriluka et al., 2014) and LSPET (Johnson & Everingham, 2011) datasets. They contain large amounts of 2D keypoint locations annotations of in-the-wild scenes, meanwhile we use their pseudo pose and shape annotations based on EFT (Joo et al., 2020).

# C  PROGRESSIVE TRAINING SCHEME

In this paper, we develop an improved progressive training scheme adapting to our model, which consists of three training phases.

**In the first phase (phase-1)**, we expect the Base model and SMPL heads to be fully trained to produce accurate estimates for static images; so we train the model only using 2D image datasets and individual frames from Human3.6M (Ionescu et al., 2013) and MPI-INF-3DHP (Mehta et al., 2017).

**In the second phase (phase-2)**, we take the pre-trained weights of Base model and SMPL heads from the phase-1 as the initialization, and then use the mixed 2D/3D and image/video datasets to train the whole model, including the Base model, Temporal Transformer and SMPL heads. In practice, we find that the generalization of the model gradually deteriorates in the middle and late periods in the second phase, even when trained with 3DPW (von Marcard et al., 2018) training set. And we further observe that the estimated pose and shape of human mesh become implausible in the later period of the training process. We empirically attribute such phenomena to that 1) a large proportion of training data has no annotations of pose and shape parameters, such as InstaVariety (Kanazawa et al., 2019b) or PoseTrack (Andriluka et al., 2018), and such data may dominate the model training

---

[2]https://github.com/rwightman/pytorch-image-models/blob/master/timm/models/resnetv2.py

[3]https://github.com/rwightman/pytorch-image-models/blob/master/timm/models/vision_transformer.py

[4]https://github.com/rwightman/pytorch-image-models/releases/download/v0.1-vitjx/jx_vit_base_resnet50_224_in21k-6f7c7740.pth

in the later period; 2) domain gap exists among various datasets; 3) we do not leverage any prior reasonable SMPL pose and shape constraints on the tokens or the estimated parameters about the pose and shape. These factors may cause the conflicts between multiple objectives in the later stage of training, such as overfittng in $\mathcal{L}_{2D}$ making the model less constrained with reasonable SMPL pose or shape parameters. Based on these observations and conjectures, we *do not use 3DPW training set* in the second phase and develop the third training phase as follow.

**In the third phase (phase-3)**, we use the pre-trained weights from the phase-2 and *fine-tune* the whole model on mixed datasets only consisting of 3DPW and Human3.6m that have accurate pose and shape annotations as supervision. This phase not only largely preserves the adaptability to in-the-wild scenarios that is learned in the phase-2, but also makes the model focus on learning to predict accurate and credible pose and shape values to fit the image appearance, 3D joint locations and 2D joint locations. Note that we do not separately fine-tune the model on each dataset and report the best performance on both datasets separately. Instead, we finally achieve a *single* best model to report the results on 3DPW and Human3.6M, which has attained wider adaptability and good generalization.

The default values of the weights $w_\theta, w_\beta, w_{norm}, w_{3D}, w_{2D}, w_{temp}$ are 60, 0.06, 1, 600, 300 and 600 unless otherwise stated. In the first phase, we use 4 Tesla V100 GPUs with a batch size of 120 for each GPU. The $w_{temp}$ is set to 0. In the second phase, we use 16 Tesla V100 GPUs to conduct distributed training (2 nodes and 8 GPUs for each node). The batch sizes for 3D video/2D video/2D image datasets are 4, 3 and 7 for each GPU. The $w_{temp}$ is set to 0. In the third phase, we use 8 Tesla V100 GPUs with a batch size of 8 for each GPU. The $w_{norm}$ is set to 0.01.

For training phase 1 and 2, we train the model for 100 epochs separately using Adam with $1e^{-4}$ initial learning rate, which decays 10 times at the 60-th and 90-th epochs. For the training phase 3, we fine-tune the model for 40 epochs using SGD with $1e^{-4}$ initial learning rate, which decays 10 times at the 20-th and 30-th epochs. We found using SGD is better than using Adam for both 3DPW and Human3.6m datasets. For data augmentation, we follow the settings in MAED (Wan et al., 2021)

Table 4: Study on the sequence length of the input video clip for the temporal Transformer model.

| Input sequence length | PA-MPJPE $\downarrow$ | MPJPE $\downarrow$ | PVE $\downarrow$ | Accel $\downarrow$ |
|---|---|---|---|---|
| T=16 | 42.2 | 76.6 | 88.8 | 18.1 |
| T=32 | 42.5 | 76.2 | 88.8 | 18.5 |
| T=64 | 42.3 | 75.9 | 88.5 | 17.4 |
| T=128 | 42.2 | 75.4 | 88.0 | 15.1 |

# D  LEARNED POSES AND EXAMPLES OF THE IN-THE-WILD AND INDOOR SCENES

In the section, to show our model indeed has learned reasonable pose using the joint rotation tokens, we also visualize the poses that are generated by sample the randomly initialized and the finally learned joint rotation tokens. We transform these tokens to the pose parameters using the learned rotation head and visualize the human meshes. As shown in Figure 9, we can see the random poses show very chaotic, distorted, and unnatural states, but the learned pose shows a relatively reasonable and natural human pose.

We compare our model with the typical video-based HMR method - VIBE (Kocabas et al., 2020) under the same detection and tracking framework provided by VIBE implementation. The reconstructed results for videos are show in Fig. 8. We encourage the readers to see the videos in the supplementary materials for better comparisons.

We show more qualitative results for some *complex scenes in the wild*, including crowded scenes, fast human motion and occluded persons. In Fig. 10, Fig. 11 and Fig. 12, we show the human mesh reconstruction results of the examples from PoseTrack (Andriluka et al., 2018) and Human3.6M (Ionescu et al., 2013). We also provide the video files on the supplementary materials. Please see the supplementary video files for more references.

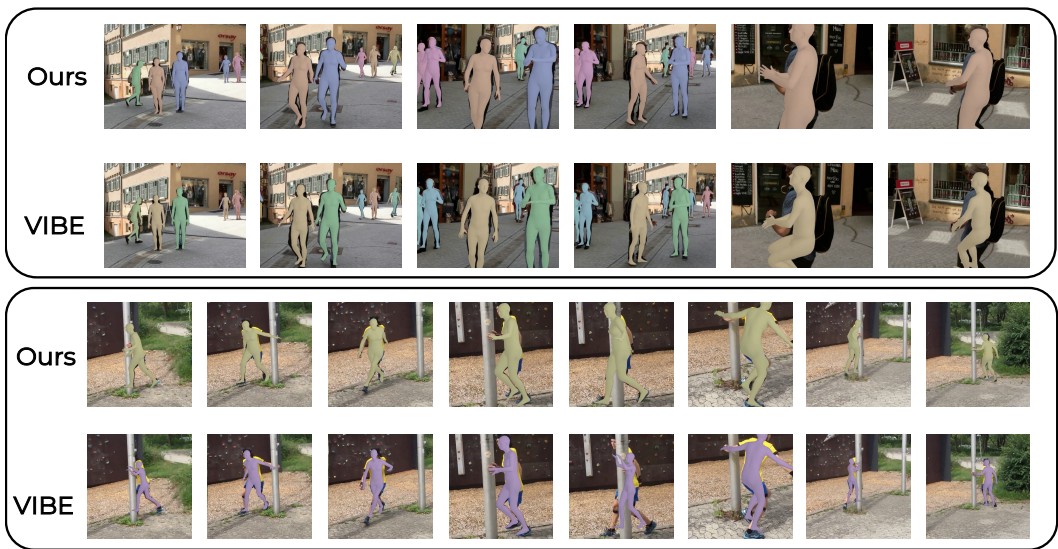

Figure 8: The qualitative comparisons between our model (top) and the VIBE model (Kocabas et al., 2020) (bottom) for two video clips in the 3DPW test set.

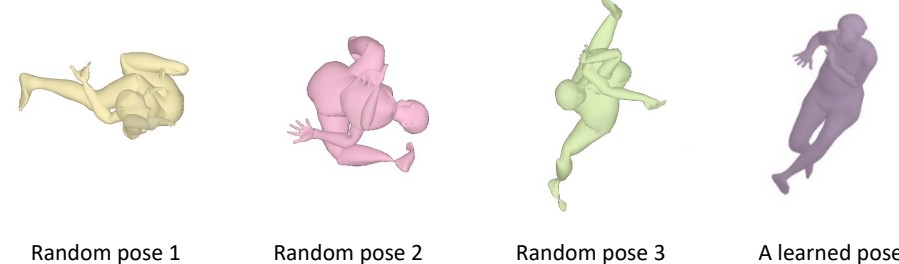

Random pose 1          Random pose 2          Random pose 3          A learned pose

Figure 9: Visual pose comparison between randomly initialized and finally learned joint rotation tokens, transformed by the learned rotation head.

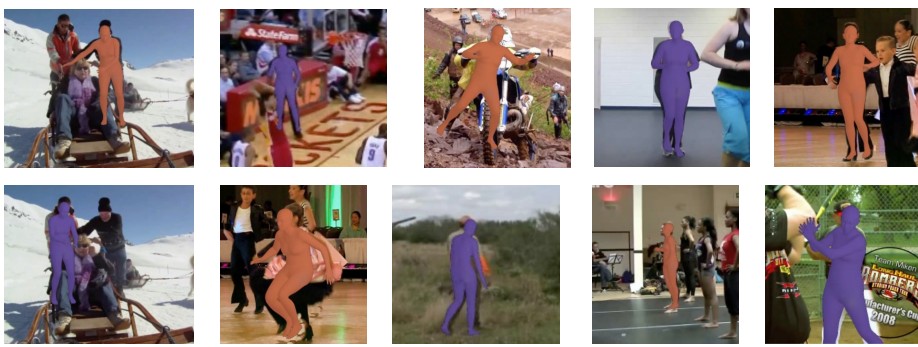

Figure 10: The human mesh reconstruction results on some hard examples from PoseTrack (Andriluka et al., 2018).

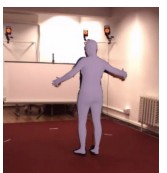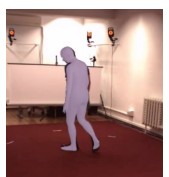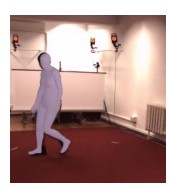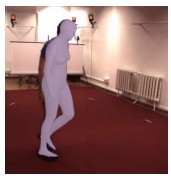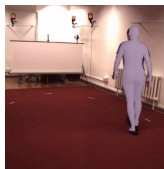

Figure 11: The human mesh reconstruction results on the examples from Human3.6M (Ionescu et al., 2013).

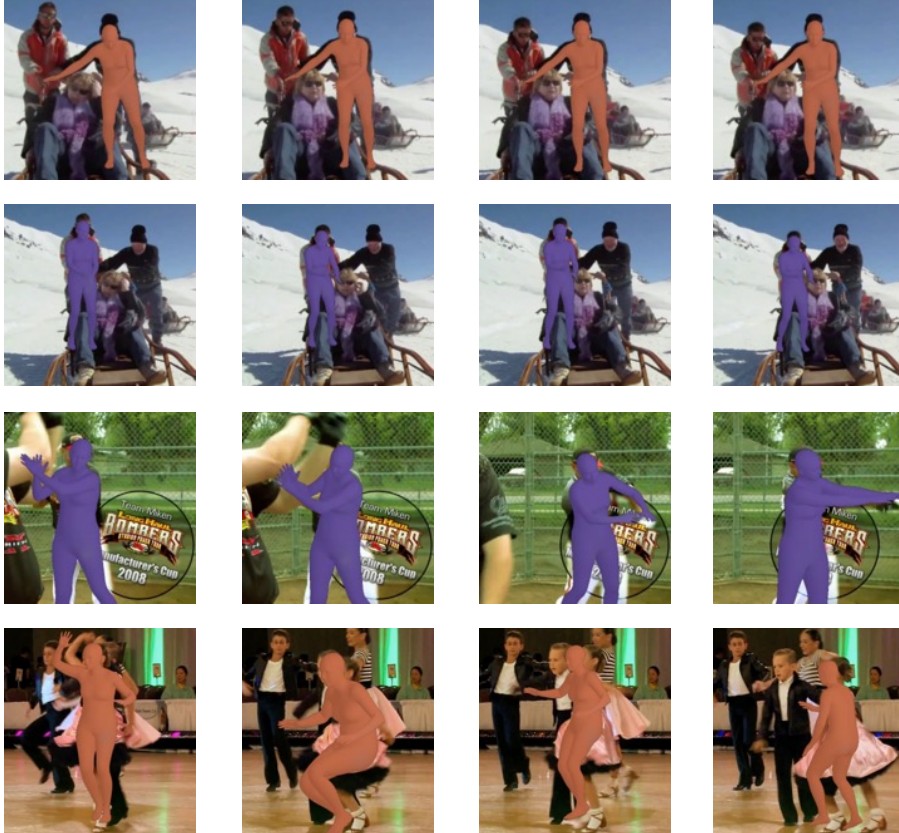

Figure 12: The human mesh reconstruction results for some video clips sampled from PoseTrack (Andriluka et al., 2018).

