# OpenReview forum: "Capturing the Motion of Every Joint: 3D Human Pose and Shape Estimation with Independent Tokens"
_ICLR.cc/2023/Conference — ICLR 2023 notable top 25%_

### Official Review · Reviewer_pLqz · 2022-10-21

**Confidence:** 4
**Clarity, Quality, Novelty And Reproducibility:** The paper is clear and seems novel.
**Correctness:** 3
**Technical Novelty And Significance:** 3
**Empirical Novelty And Significance:** 3
**Recommendation:** 6

**Strength And Weaknesses:**

Strengths
1. Learning the prior poses and shape in a data-driven way using self-attention is an interesting point.
2. Strong performance on multiple benchmarks.

Weaknesses

1. Insufficient ablation studies.
One major contribution of this paper is that it learns prior poses and shapes through the self-attention mechanism. However, there is no direct comparison between 1) using fixed average poses and shapes (used in HMR) and 2) using learnable poses and shapes, while fixing all other components of the system. This ablation is necessary to show the effectiveness of the learned pose and shape priors.

2. Lack of justification of the architecture.
In Fig. 2, there can be several variants. First, heads can additionally take image feature tokens instead of only taking prior tokens. Second, passing only image feature tokens (without prior tokens) to Transformer and passing prior tokens to the heads following iterative feedback-based methods (HMR and others). Basically, I can’t get fundamental design philosophy of the current system that takes prior tokens and uses features only from prior tokens for the final output.

3. Temporal modeling only with a single joint might be sub-optimal as the output rotations are relative rotation wrt. their parent joints. Hence, I think three joints, including child and parent joints, might perform better.


**Summary Of The Paper:**

This paper presents a 3D human pose and shape estimation system that takes a RGB video as an input. Most of the previous works take pre-defined mean poses and shapes for the iterative feedback-based regression. On the other hand, the proposed system learns such prior poses and shapes through the self-attention mechanism. In addition, the proposed system models temporal information with separate joint-level tokens. Experimental results demonstrate powerful performance of the system.

**Summary Of The Review:**

Although the paper proposes some interesting improvements, lack of ablation study and justification of the design are the weaknesses.

-- post-rebuttal comments--
The authors somehow resolved part of my concerns, but they did not provide experimental results in an quantitative way. Therefore, I'd like to keep my original score instead of raising it.

---

> ### Author Response · Authors · 2022-11-11
> **Author response to the Reviewer pLqz (1/2)**
>
> We thank you for your thoughtful comments and valuable feedback. We hope that our response can address your concerns:
>
> > ***Q1: Insufficient ablation studies***
>
> **A1**: Thanks for your suggestions. As a matter of fact, we attempted to introduce the mean parameters to this model for comparison. But, in practice, it’s hard to perform such an ablation study that incorporates the mean pose and shape parameters in this framework, because we don’t regress parameters in an iterative error feedback manner. If we use the mean parameters as initialization, the model needs to learn a linear mapping from the rotation 6-dim parameters to the 764-dim token vector, as well as the 10-dim shape parameters to the 764-dim token vector. But the initialized weights of this linear layer are totally random, making the input mean parameters of pose and shape ineffective.
>
> For another case, if we use the mean pose and shape as initialization and an iterative regressor to take as input the feature vector outputted from the backbone, it will fundamentally change the model and become a new design. This experiment has been validated in MAED [1]. As shown in Table 2 of the MAED paper, the CNN+STE+Iterative model based on the iterative regression obtains 47.5 mm for PA-MPJPE, which is worse than the result (45.7 mm) of the MAED model (CNN+STE+STE). Since we use the same backbone and the same training data as MAED, it can be said that our model (42.0 mm) is better than MAED and the iterative regressor scheme using the mean pose and shape initialization.
>
> As for your second point, we show the learnable pose and shape while fixing the other components in Figure 5 (a). We visualize the learnable prior pose and shape, by sending the learned joint rotation tokens and shape tokens to the rotation head and shape head. Note that, these tokens are randomly initialized before being trained, and finally learn reasonable human pose and shape priors, as shown by the visualized human appearance (Figure 5(a)). For better comparison, we have added a new Figure 9 to the appendix of the paper. There, we show some randomly initialized poses that are transformed from the randomly initialized joint rotation tokens. We can see the random poses take on very chaotic, distorted, and unnatural states, but the learned pose shows a reasonable and natural human appearance. Please see the revision. We have conducted several key ablation studies in the current work and we will conduct more experiments in the future.
>
> > ***Q2: Lack of justification of the architecture***
>
> **A2**: Yes, there may be many architecture variants to predict the final shape, pose, and camera parameters. A straightforward approach is to pass the CNN image feature tokens to the Transformer and use an iterative feedback regression method like HMR [2]. But this approach is not essentially different from the previous HMR methods. In this case the Transformer is only used to enhance the ability to extract features. In fact, this scheme has been verified in the paper of MAED (which we mentioned in A1), where the authors designed the CNN+STE+Iterative scheme as shown in Table 2 of their paper.
>
> The fundamental design philosophy of our model indeed differs from previous HMR methods. The motivation is that the human body's 3D joint rotations and shape are not completely dependent on the image features and they can be abstracted from image pixels as independent concepts/representations. The relationships among joint rotations and body shape have their prior states $P(\Theta)$. Given an observed image $I$ as a condition, the posterior estimate $P(\Theta | I)$ about the pose and shape parameters can be inferred by updating the tokens.
>
> In addition, the SMPL heads we use are not MLPs but linear layers. The rotation head builds a fixed linear mapping between the joint rotation token vector (764-dim) and the joint rotation parameter (6-dim). The shape head builds a fixed linear mapping between the shape token (764-dim) and shape parameters (10-dim). Thus, these tokens have strong linear relationships with the joint 3d rotation or shape parameter values. Benefiting from this, the model can capture the rotational temporal motion at the joint level, which is infeasible for conventional HMR methods.
>
> We embody these ideas into the model design, which constitutes a novel architecture and has been recognized by the reviewer EHtd and w5go as well. The results show our simple architecture achieves better performance than the previous HMR-based methods and video approaches, which also demonstrates the effectiveness and superiority of this design.

---

> > ### Author Response · Authors · 2022-11-11
> > **Author response to the Reviewer pLqz (2/2)**
> >
> > > ***Q3: Temporal modeling only with a single joint might be sub-optimal***
> >
> > **A3**: For the 3D spatial location, the 3D position of a joint indeed has strong relevance with its parent joint or its child joints due to the tree-based human kinematics. But for the 3d rotations of joints, it’s unclear if they have determined parent-joint dependencies. Therefore, we make no assumptions about directed dependencies in a kinematic tree between the joint rotations, but expect the model to learn these mutual relationships from data using self-attention mechanism. In this learning framework, parent-joint dependencies and *non-adjacent yet important dependencies* can be learned. In the temporal modeling, the self-relevance in the rotation for a joint is more pronounced, because the 3D rotation of a specific joint may change continuously or repeat over time when a person moves or performs certain actions. Thus, the joint 3D rotation estimation for the current frame can be regularized by using the joint information from its past and future states. Based on these assumptions, we design the temporal model.
> >
> > While, our model doesn’t abandon the parent-child dependency between joints. It learns the intrinsic spatial cues and parent-child dependencies in the base model part. In the temporal modeling stage, the model can focus on modeling the joint rotation over time. It is worth noting that MAED adopts a tree-based kinematic topology decoder and uses spatial-temporal parallel modeling. In comparison to this scheme, we make the ablations for comparison. As shown in Table 3 of the paper, our temporal modeling shows advantages over the proposed temporal scheme in MAED.
> >
> > **References:**
> >
> > [1] Wan et al. “Encoder-decoder with multi-level attention for 3d human shape and pose estimation”, ICCV 2021
> >
> > [2] Kanazawa et al. “End-to-end recovery of human shape and pose”, CVPR 2018

---

> > > ### Comment · Reviewer_pLqz · 2022-11-15
> > > **Discussion**
> > >
> > > Hi, thanks for your responses. I'd like some questions.
> > >
> > > 1. For the W1, the authors said that Fig.5 (a) shows learned pose and shape priors, but I can't clearly get. How the tokens are converted to SMPL rotations and shape parameters without image features? Fig.2 shows that image features should be passed to recover SMPL parameters. Did you pass zero image features to Transformer blocks to visualize the learned pose and shape priors?
> > >
> > > 2. For the W2, the authors explained a concept of prior and posterior, which are interesting for me. However, one concern is that current system does not model some joint distribution of image features and all joints as all tokens are processed with shared layers in Transformer blocks. As each human joint has its own characteristic, using joint-specific weight is proven to be beneficial than using a shared weight in [A]. Also, I'm not sure estimating SMPL parameters only from prior tokens at the SMPL head is an optimal one as the tokens could not capture enough posterior information. Providing additional image features to SMPL layers can capture posterior I guess?
> > >
> > > [A] Liu, Kenkun, et al. "A comprehensive study of weight sharing in graph networks for 3d human pose estimation." ECCV, 2020.
> > >
> > > 3. For the W3, the authors said that the parent-child dependency might be weak for 3D joint rotations, which I do not agree with. The 3D position is a global representation, where global means relative to the origin. On the other hand, 3D rotations are local representation, where local means relative to the parent joint. Hence, when we recover 3D joint positions from 3D rotation using forward kinematics, 3D rotations of all ancestor joints are considered. Although joint-level features can contain some parent/child joint information, I think using more joints could produce better results. This kind of joint cluster, which consists of parent/itself/child joints, has been used in many 3D human modeling works, such as [B]
> > >
> > > [B] Osman, Ahmed AA, Timo Bolkart, and Michael J. Black. "Star: Sparse trained articulated human body regressor." ECCV, 2020.
> > >
> > > 4. By the way, the authors did not cite [C], which introduced joint-level features for more accurate whole-body/body/hand 3D pose estimation. Please cite this work.
> > >
> > > [C] Moon, Gyeongsik, Hongsuk Choi, and Kyoung Mu Lee. "Accurate 3D Hand Pose Estimation for Whole-Body 3D Human Mesh Estimation." Proceedings of the IEEE/CVF Conference on Computer Vision and Pattern Recognition. 2022.

---

> > > > ### Author Response · Authors · 2022-11-15
> > > > **Author response to the Reviewer pLqz**
> > > >
> > > > Thanks for your questions and comments. We hope the following clarifications can help address some of the concerns.
> > > >
> > > > **A1**: Thanks for this question. To show Fig.5 (a), we directly pass 24 prior joint rotation tokens to the linear rotation head, and pass the prior shape token to the linear shape head, not following the pipeline in Fig.2. This rotation head has learned to convert a 764-dim joint rotation token to the rotation-6d representation. The shape head has learned to convert a 764-dim shape token to the 10-dim shape parameter. Then we convert the rotation 6d representation of each joint to its rotation matrix and visualize the SMPL pose. The tokens used to visualize prior pose and shape are not the tokens directly sent to the rotation head and shape head in Fig.2, but the initial ones annotated with “prior tokens” in Fig.2, without using image feature information, nor passing zero image feature.
> > > >
> > > > **A2**: In this framework, we assume the prior tokens can learn to encode the prior distribution of the pose and shape. When we send the image feature tokens and prior tokens to the Transformer, the joint distribution of image features, joint rotations, shape and camera parameters can be uniformly modeled by the Transformer. Please note that the joint rotation tokens, shape token and camera token are updated progressively through multiple transformer layers. The tokens used to estimate SMPL parameters are no longer the initial prior tokens, but are updated when interacting with image features through self-attention layers. Therefore, we can say that the final tokens directly for estimating pose and shape parameters have captured the posterior information of the image. From another perspective, the final transformer layer acts like the final iteration of the MLP regressor in HMR.
> > > >
> > > > We thank you for mentioning [A]. It provides a good insight that using joint-specific weight is beneficial than using a shared weight. We also agree that not all joints are directly related and some may have negative mutual effects. In this framework, we leave the learning of these coupled or decoupled relationships to the self-attention mechanism, which can adaptively adjust the relationship between joints through attention weights.
> > > >
> > > > [A] Liu, Kenkun, et al. "A comprehensive study of weight sharing in graph networks for 3d human pose estimation." ECCV, 2020.
> > > >
> > > > **A3**:  We completely agree with you that 3D rotations are local representations relative to the parent joints and the 3D joint positions can be computed from 3D rotations using forward kinematics following their ancestor joints in a tree. But this doesn't mean that the 3d rotation value of a joint strongly depends on the rotation value of its parent or its ancestors. The 3D positions do have such *directed dependencies* due to the forward kinematics, but the 3D rotations may not have such obvious dependencies. We don't know their relationships, and thus make no assumptions about directed dependencies on the joint rotations. As you said, we also think using more joints is possibly better. Considering it's difficult for us to design a hand-crafted strategy, instead we adopt the self-attention mechanism to expect the base model to learn these unknown mutual relationships from data.
> > > >
> > > > **A4**: We thank you for mentioning the paper [C] for reference. We will add this paper for comparison and reference in the revision.
> > > >
> > > > [C] Moon, Gyeongsik, Hongsuk Choi, and Kyoung Mu Lee. "Accurate 3D Hand Pose Estimation for Whole-Body 3D Human Mesh Estimation." Proceedings of the IEEE/CVF Conference on Computer Vision and Pattern Recognition. 2022.

---

> > > > > ### Comment · Reviewer_pLqz · 2022-11-16
> > > > > **Discussion**
> > > > >
> > > > > Thanks for the authors' detailed responses. Regarding A2 and A3, I think authors' responses can be reasonable. But it would be more solid if the authors can provide experimental results with 3D error changes. I think this paper can give good insights to following papers, and if there are many ablation studies including my suggestions, the paper can become much stronger.

---

> > > > > > ### Author Response · Authors · 2022-11-17
> > > > > > **Discussion**
> > > > > >
> > > > > > Thanks for your suggestions. We would conduct more experiments and ablations in the future as you suggested.

---

### Official Review · Reviewer_w5go · 2022-10-21

**Confidence:** 3
**Correctness:** 4
**Technical Novelty And Significance:** 2
**Empirical Novelty And Significance:** 2
**Recommendation:** 6

**Clarity, Quality, Novelty And Reproducibility:**

- The paper is clearly written and is easily read with its details.
- The overall experiments are well designed and uses various datasets to conduct quantitative evals.
- Adding extra tokens are not novel, but their modeling of individual joint rotations, and combining with temporal Transformer is shown performing well.
- The paper has enough detail information to reproduce.


**Strength And Weaknesses:**

Strengths:
- the overall performance improved, and achieves SOTA on various metrics.
- ablation studies are well performed.

Weaknesses:
- camera models are simplified, so it's unclear how it will more generally perform under severe camera distortions.
- sensitivity on initial human detection for clipping the region was not tested.
- it is unclear why ACCEL performance is not improved compared to others.

**Summary Of The Paper:**

The paper introduces a Transformer-based method for estimating 3D human pose and shape, and achieves improved performances over SOTA. They adapt previously known visual Transformer architecture overall and extends to add new independent tokens to encode joint rotation, shape, and camera parameters into the network. The outcome performance is improved for various metrics, and qualitatively reduces jitters of the motion.

**Summary Of The Review:**

The paper shows good performance gains over existing approaches. Their modeling of the 3D human pose and shape estimation network is sounding, and adding temporal Transformer is showing positive quality gain as well. There are a few questions and weak points as described above, but overall the paper was worth reading and provides good insight.

---

> ### Author Response · Authors · 2022-11-11
> **Author response to the Reviewer w5go**
>
> We are very grateful to the reviewer for the positive comments and valuable feedback. We hope that our response can address your concerns.
>
> > ***Q1: About the camera model***
>
> **A1**: In this work, following the common HMR methods, our model is still based on some simple camera assumptions, including a weak-perspective projection camera model, constant focal length, and zero camera rotation relative to the world coordinate frame. And it's difficult to evaluate the model performance under conditions of severe camera distortions. In this paper, we focus on model architecture design and temporal modeling. The camera model assumption is not within the scope of this work and the architecture is also not coupled with the camera model used. This means that we can incorporate a more reasonable camera model, such as a perspective projection model, into this framework to improve performance. What’s more, we have a camera token for end-to-end learning, so it is possible to estimate more accurate camera parameters, such as SPEC [1], for better human reconstruction and projection.
>
> > ***Q2: sensitivity on initial human detection for clipping the region was not tested.***
>
> **A2**: Thank you for your reminder and suggestion. In this work, the sensitivity problem is not considered as a major issue. In practice, we enlarge the detected bbox with a scale factor to crop the human region. The scale factor is set to 1.3 during training. For inference, we found a small improvement using a scale factor of 1.1 over using 1.2 and 1.3. However, from the effect of the video, due to the existing deviation and jitters of the detected human bounding boxes for the consecutive frames, our method can still maintain a relatively robust mesh prediction for the video input.
>
>
> > ***Q3: The performance improvement on ACCEL error.***
>
> **A3**: ACCEL error computes the average difference between ground truth 3D acceleration and predicted 3D acceleration of each joint in mm/s. The GT acceleration is computed by $(p_{t+1}-p_t) - (p_t- p_{t-1})$, where $p_t$ is the 3D position of a joint at timestamp $t$. This error can measure smoothness, but it's not a stable indicator. The overall shifts on the predicted 3D position may still have a low Accel error, while the mesh may not be aligned with the image. It may be necessary to combine the PA-MPJPE and MPJPE metrics for overall performance evaluation. In Table 2, we can see that our temporal modeling improves on both the PA-MPJPE and Accel metrics. For the TCMR [2] and MPS-Net [3], they remove the residual connections between the static and temporal features, decreasing the Accel error significantly. Compared to them, our model lags in the accel error but leads on all other metrics. And we also observed the differences in video results between theirs and our approach. Our model produces better pixel-alignment meshes and natural human motions. When compared with the state-of-the-art temporal approach – MAED [4], our model still has obvious advantages on the Accel and PA-MPJPE errors.
>
> **References:**
>
> [1]: Kocabas et al. “SPEC: Seeing People in the Wild with an Estimated Camera”，ICCV 2021
>
> [2]: Choi et al. “Beyond static features for temporally consistent 3d human pose and shape from a video”, CVPR 2021
>
> [3]: Wei et al. “Capturing humans in motion: Temporal-attentive 3d human pose and shape estimation from monocular video”, CVPR 2022
>
> [4]: Wan et al. “Encoder-decoder with multi-level attention for 3d human shape and pose estimation”, ICCV 2021

---

### Official Review · Reviewer_EHtd · 2022-10-24

**Confidence:** 4
**Correctness:** 4
**Technical Novelty And Significance:** 3
**Empirical Novelty And Significance:** 3
**Recommendation:** 8

**Clarity, Quality, Novelty And Reproducibility:**

In my opinion the clarity of presentation is fine given the space constraints. To the best of my knowledge, the presented approach is novel for the task at hand. Transformer architectures have been applied before for the problem, but this specific approach allows for state-of-the-art results and temporal consistency. Similarly, independent tokens have been proposed before, but not in the given context. Furthermore, the use of these tokens allows for a straightforward temporal extension, which constitutes a novel contribution to the best of my knowledge. Lastly, reproducibility is not an issue here to the best of my understanding.

**Strength And Weaknesses:**

The manuscript clearly presents the core idea and provides intuitive justification on why the proposed approach is beneficial for the task at hand. The literature overview is comprehensive within the space limitations, the method description is clear, and the experimental results are extensive and convincing. Apart from a few syntax and grammar errors (and also the use of the word "achieve" instead of "compute", please fix this) this is a well-written paper overall.

I don't have any major points of criticism, rather some thoughts for potential improvement: Recent advances in human pose estimation using transformers indicate that it is beneficial to use more queries than the total number of estimated keypoints (see eg Li et al. "Pose Recognition With Cascade Transformers" 2021 -- a good reference to add to the literature overview by the way, despite the fact that it only regresses 2D keypoints). It is not clear whether such an approach would also be beneficial with the proposed approach, or even possible. Please consider commenting on this, it is unclear to me whether it also makes sense to experiment on this.

**Summary Of The Paper:**

This paper proposes an approach for the estimation of human body 3D pose and shape from monocular input (single image or video). The core idea is to use learnable independent tokens for the joint rotations, the camera parameters (the weak-perspective camera model is used) and the body shape, as expressed in the used SMPL model. This allows for an extension for the case of video, with the temporal information first captured per joint by the respective token, and then decoded by a suitable temporal transformer. This approach implicitly imposes temporal smoothness, yielding temporally consistent results with minimized (but not eliminated) jitter.

**Summary Of The Review:**

Overall I think that, despite the limited novelty, it is still a well presented idea that achieves state-of-the-art results in the problem that is worth accepting in ICLR.

---

> ### Author Response · Authors · 2022-11-10
> **Author response to the Reviewer EHtd**
>
> We appreciate your thoughtful comments and valuable suggestions. We are encouraged by your positive comments on this work. We hope that our response can address your confusion.
>
> > ***Q1: Comments on the potential improvements if using more queries like PRTR***
>
> **A1**: We thank you for mentioning the PRTR [1] model and your insights. We have added this paper for reference in the revision. PRTR follows the query design philosophy in DETR [2], which means each query doesn’t have a fixed correspondence to the type of keypoint. Each query can be responsible for predicting any keypoint class. And the results show that it’s better to make the number of queries larger than the number of types of human keypoints. This phenomenon is interesting and can be explained.
> As the final queries need to predict the spatial location of different types of keypoints, they need to match the groundtruth keypoints (visible) to receive supervision during training. As a result, for a given person instance, some queries may be matched and some may not be matched. Due to the data distribution of keypoints in the training dataset, each query finally has learned its own preference in spatial areas and keypoint classes, which is also revealed in Fig.7 of the DETR paper and Fig.6 of the PRTR paper. After training, each query embedding probably has learned its priority and has its spatial distribution and class preferences. Therefore, using redundant queries would be beneficial for matching keypoints that are possibly distributed anywhere.
>
> Different from their designs in query, we set the number of joint tokens as the total number of joints, each of which corresponds to a certain joint type. It is inspired by the CLS token of ViT [3] and the keypoint tokens of TokenPose [4]. This scheme is reasonable for 3D human reconstruction because we need to predict the 3d rotation rather than the 2D image spatial location of each joint. The 3d joint rotation has no spatial distribution but has a certain value in the rotation space, so we don’t need the matching process. Furthermore, a joint may be invisible in the image space and hard to predict; but each joint rotation exists independent of its image visibility. There is no extra need for more redundant queries to match the joint rotations of a human instance, as we use a complete SMPL human pose model to fit the human body in an image.
>
> As for this 3D prediction task, the benefit of fixing the correspondences is that the explicit relationships among these joints can be learned. If we use the PRTR’s scheme that each token/query has no explicit semantics of a particular joint, the model may be unable to learn prior relationships between joints. Although it’s feasible to introduce the matching mechanism and classification head, it would induce another completely new model and increase the redundancy of the current system. We don’t ensure that it will be suitable for this 3D human reconstruction task using a parametric model.
>
> > ***Q2: Fixing the syntax and grammar errors***
>
> **A2**: Thanks for your suggestions. We have fixed the use of the word “achieve” and carefully checked the main text again to fix all syntax and grammar errors.
>
>
> **References:**
>
> [1] Li et al. "Pose Recognition With Cascade Transformers, CVPR 2021
>
> [2] Carion et al. “End-to-end object detection with transformers”, ECCV 2020
>
> [3] Dosovitskiy et al. “An image is worth 16x16 words: Transformers for image recognition at scale”, ICLR 2020
>
> [4] Li et al. “TokenPose: learning keypoint tokens for human pose estimation”, ICCV 2021

---

### Decision · Program_Chairs · 2023-01-20

**Decision:**

Accept: notable-top-25%

**Justification For Why Not Higher Score:**

The proposed method is on a fairly specific application of 3D human shape and pose prediction, which is of interest to a subset of the community.

**Justification For Why Not Lower Score:**

The idea of modelling 3D priors as independent token embeddings, which is then combined with the image to obtain an image-conditioned embedding, can be useful for potentially other tasks.

**Metareview: Summary, Strengths And Weaknesses:**

Summary: The paper proposes to estimate 3D human pose and shape from monocular input (either a single image or a video) by modelling the joint rotations, body shape, and camera using learnable token embeddings that capture prior knowledge about human shape and pose.   These tokens are "independent" from the image features, and are learned from large amounts of training data.  These tokens combined with the image features are used as input to a transformer-based architecture which transforms the independent token embeddings into image-conditioned embeddings.  These image-conditioned embeddings then are used predict the final camera parameters and human pose and shape (in the form of SMPL parameters).  A temporal transformer is used to enforce temporal consistency.  Experiments show that the proposed method outperforms prior methods.

Strengths:
- The idea of modelling the priors for the human pose and shape as learnable independent tokens is novel and can potentially be useful for other tasks as well
- Experiments are in general well-designed and show the effectiveness of the proposed method
- Reviewers found the paper to be clearly written and easy to read, giving a clear intuition and justification for the approach

Weaknesses:
- There is limited ablation studies and justification of specific architecture choices
- Some of the modelling is simplified and can be improved (e.g. the temporal modelling of joints, camera model)


**Note From Pc:**

if the above contains the word "oral" or "spotlight" please see: "oral" presentation means -> notable-top-5% and "spotlight" means -> notable-top-25%. As stated in our emails, we are disassociating presentation type from AC recommendations